# DeepMath-103K: A Large-Scale, Challenging, Decontaminated, and Verifiable Mathematical Dataset for Advancing Reasoning

**Zhiwei He**[*,1,2] **Tian Liang**[*,1] **Jiahao Xu**[*,1] **Qiuzhi Liu**[1] **Xingyu Chen**[1,2] **Yue Wang**[1]
**Linfeng Song**[1] **Dian Yu**[1] **Zhenwen Liang**[1] **Wenxuan Wang**[1] **Zhuosheng Zhang**[2]
**Rui Wang**[†,2] **Zhaopeng Tu**[†,1] **Haitao Mi**[1] **Dong Yu**[1]

[1]Tencent   [2]Shanghai Jiao Tong University

 https://github.com/zwhe99/DeepMath

 https://hf.co/datasets/zwhe99/DeepMath-103K

## Abstract

Reinforcement learning (RL) with large language models shows promise in complex reasoning. However, its progress is hindered by the lack of large-scale training data that is sufficiently challenging, contamination-free and verifiable. To solve this problem, we introduce **DeepMath-103K**, a large-scale mathematical dataset designed with high difficulty (primarily levels 5-9), rigorous decontamination against numerous benchmarks, and verifiable answers for rule-based RL reward. It further includes three distinct R1 solutions adaptable for diverse training paradigms such as supervised fine-tuning. Spanning a wide range of mathematical topics, DeepMath-103K fosters the development of generalizable and advancing reasoning. Notably, models trained on DeepMath-103K achieve leading results on challenging mathematical benchmarks and demonstrate generalization beyond math such as biology, physics and chemistry, underscoring its broad efficacy.

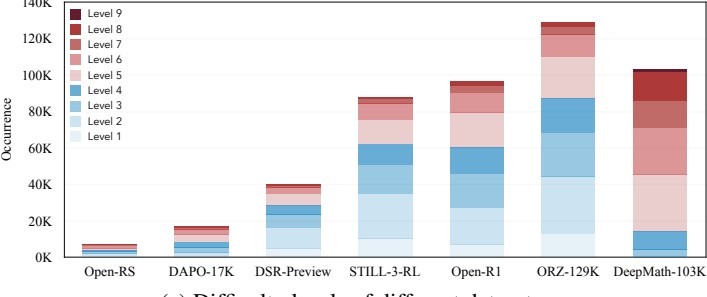

(a) Difficulty levels of different datasets.

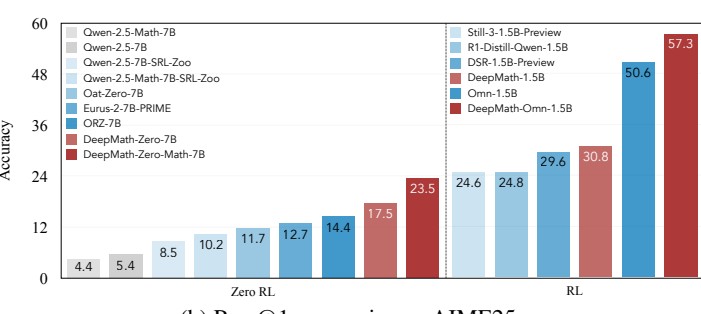

(b) Pass@1 accuracies on AIME25.

Figure 1: (a) DeepMath-103K is challenging compared to existing datasets. (b) Results of DeepMath series models under zero RL and RL setting using DeepMath-103K.

---

*Equal Contribution. The work was done when Zhiwei, Xingyu, and Yue were interning at Tencent.

†Correspondence to: Zhaopeng Tu <zptu@tencent.com>  and Rui Wang <wangrui12@sjtu.edu.cn>.

## 1 INTRODUCTION

Reinforcement learning (RL) with large language models (LLMs) has demonstrated significant potential in complex mathematical reasoning (Guo et al., 2025; Hu et al., 2025; Zeng et al., 2025a; Liu et al., 2025). Despite this promise, the effective advancement of RL is constrained by existing training data. While numerous datasets are available, they fall short in several key aspects crucial for training advanced reasoning models: (1) insufficient difficulty (Figure 1a) to push the boundaries of current models (Dang & Ngo, 2025; Yu et al., 2025; Luo et al., 2025; Face, 2025; Hu et al., 2025), (2) contamination with standard benchmarks (appendix B), (3) a lack of verifiable answers essential for RL with verifiable rewards (RLVR) (Guo et al., 2025; Cobbe et al., 2021; Hendrycks et al., 2021b; Yu et al., 2024), or (4) an inadequate combination of these critical aspects at scale. Furthermore, many of existing datasets represent the recombination and filtration of common sources (such as AIME (MAA, a)) which contain already well-formatted data, thus lacking a substantial influx of novel and diverse problems from more varied but less structured sources (Dang & Ngo, 2025; Yu et al., 2025; Luo et al., 2025; Face, 2025; Hu et al., 2025).

To bridge this gap, we introduce **DeepMath-103K**, a large-scale mathematical dataset tailored for advancing reasoning via RLVR. DeepMath-103K distinguishes itself through several key features.

- **Challenging Problems:** DeepMath-103K features a high concentration of challenging mathematical problems, with a difficulty distribution skewed towards higher levels ($\geqslant 5$) compared to existing open resources (Figure 1a).

- **Rigorous Decontamination:** To ensure trustworthy evaluation, DeepMath-103K underwent a rigorous decontamination process against a comprehensive suite of benchmarks.

- **Verifiable Answers and Diverse Solutions:** To enable rule-based reward functions in RLVR, every problem in DeepMath-103K includes a verifiable final answer that has been validated for easy extraction and verification via rules. Each problem is further enriched with three distinct R1 solutions (Guo et al., 2025), supporting diverse training paradigms such as SFT.

Beyond these core features, DeepMath-103K also differentiates itself in its raw data acquisition methodology. The prevalent trend in existing open datasets often recombines readily available and well-formatted problems from common sources such as AIME (MAA, a). This approach does not create new problems, but re-collect existing ones, which leads to significant overlaps among different datasets. Recognizing the potential limitations and eventual exhaustion of common resources, DeepMath-103K draws its content from more diverse but less structured sources, notably including discussions from Math StackExchange[1]. The raw content from these sources is informal discourse and lacking a standard format. After a rigorous curation pipeline that transformed these discussions into a well-structured QA format, DeepMath-103K is characterized by its unique problem variety and diversity compared to existing datasets.

Consequently, models trained on DeepMath-103K achieve leading results (Figure 1b):

- **Zero RL Training**: Starting from the Qwen-2.5-(Math)-7B (Team, 2024), DeepMath-Zero-(Math)-7B shows pass@1 improvements of +12.7 (+23.0) on AIME24 and +12.1 (+19.1) on AIME25, surpassing other baselines.

- **RL Training**: Initialized from instruction-tuned models, DeepMath variants also show substantial gains. DeepMath-1.5B, starting from R1-Distill-Qwen-1.5B (Guo et al., 2025), achieves pass@1 accuracy improvements of +7.9 on AIME24 and +6.0 AIME25. DeepMath-Omn-1.5B, built upon OpenMath-Nemotron-1.5B (Moshkov et al., 2025), reaches pass@1 accuracies of 64.0 on AIME24 and 57.3 on AIME25, surpassing o1-mini (63.6 on AIME24) and low effort o3-mini (60.0 on AIME24).

- **Generalizable Reasoning beyond Math**: DeepMath series models also generalizes their reasoning abilities to broader domains, achieving the best GPQA-Diamond (biology, physics and chemistry) (Rein et al., 2024), MMLU-STEM (science and engineering reasoning) (Hendrycks et al., 2021a) and BBH (complex multi-step reasoning) (Suzgun et al., 2022) scores compared to the baselines.

---

[1] https://math.stackexchange.com

These results underscore the value of DeepMath-103K as a resource for developing advanced reasoning models with broad applicability. To foster future research, we have released the DeepMath-103K dataset, along with the code and model weights, hoping to enable further exploration of advanced reasoning techniques and the development of robust and generalizable machine intelligence.

## 2 OVERVIEW OF DEEPMATH-103K

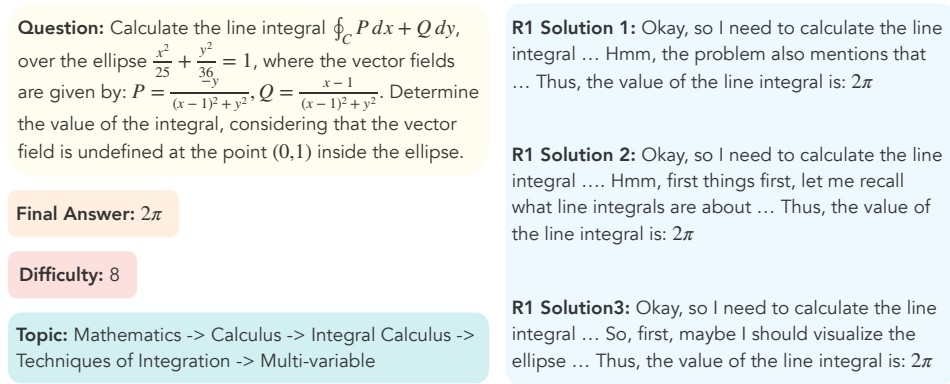

Figure 2: A data sample from DeepMath-103K.

Each data sample in DeepMath-103K is intentionally structured to be comprehensive, supporting a variety of downstream applications in mathematical reasoning research. As illustrated in Figure 2, a single sample includes the following components:

- *Question*: The mathematical problem statement.
- *Final Answer*: A verifiable final answer, crucial for rule-based reward functions in RLVR.
- *Difficulty*: A numerical difficulty score, which facilitates techniques like difficulty-aware training (e.g., curriculum learning) or adaptive compute allocation based on problem complexity (Wang et al., 2025b; Chen et al., 2024).
- *Topic*: A hierarchical topic classification for the problem, enabling topic-specific analysis.
- *R1 Solutions*: Three distinct reasoning paths generated by the DeepSeek-R1 model (Guo et al., 2025), suitable for diverse training paradigms such as SFT.

DeepMath-103K possesses several key characteristics that make it particularly suitable for advancing mathematical reasoning research:

**Higher Difficulty** DeepMath-103K includes mathematical problems spanning difficulty levels 3 through 9. The core of the dataset consists of 95K challenging problems (levels 5-9) specifically curated for this research. To ensure broader difficulty coverage, this dataset is augmented with an additional 8K problems (levels 3-5) sourced from SimpleRL (Zeng et al., 2025b). For comparison, we analyzed and labeled the difficulty levels of several existing datasets commonly used for RLVR training in math domain: Open-RS (Dang & Ngo, 2025), DAPO-17K (Yu et al., 2025), DSR-Preview (Luo et al., 2025), SITLL-3-RL (Chen et al., 2025), ORZ-129K (Hu et al., 2025), and Open-R1 (Face, 2025). Figure 1a illustrates the difficulty distributions across these datasets. As depicted, DeepMath-103K exhibits a significantly more challenging problem distribution, containing a substantially higher proportion of problems at difficulty level 5 and above compared to the other benchmark datasets. This focus on higher difficulty is intended to push the reasoning limits of current models.

**Rigorous Data Decontamination** DeepMath-103K was constructed exclusively using the training splits of existing open resources, with careful avoidance of any known test set materials. However, our preliminary analysis revealed that these source data exhibits alarmingly high levels of contamination with problems from commonly used evaluation benchmarks. As illustrated in Figure 3, the

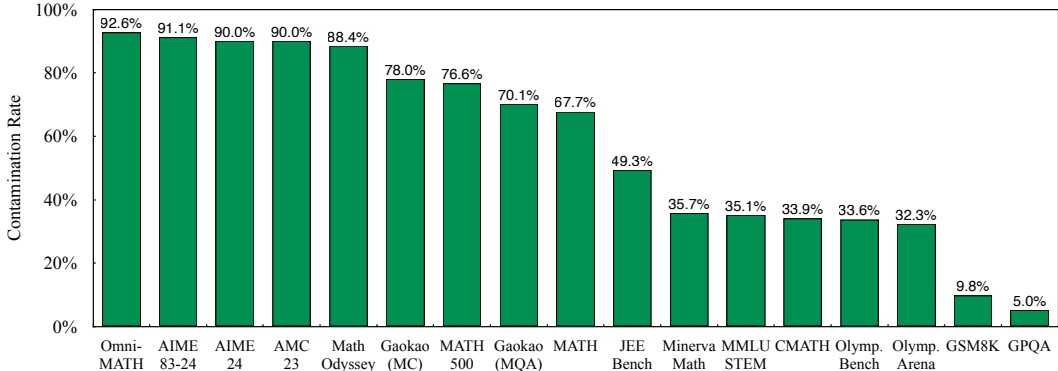

Figure 3: Contamination rates of common mathematical and STEM benchmarks detected in the raw data sources before decontamination.

contamination rates (defined as the percentage of benchmark test samples found within our raw data pool) are notably high: reaching 90% for AIME24 and AMC23, 76.6% for MATH500, 35.7% for Minerva Math, and 33.6% for OlympiadBench. Recognizing that these benchmarks are frequently employed for model evaluation, DeepMath-103K underwent a rigorous decontamination procedure. This process systematically identified and removed problems that overlap with these standard evaluation sets, ensuring the integrity and reliability of future benchmark results obtained using models trained on DeepMath-103K.

**Broad Topical Coverage** Complementing its high difficulty and data integrity, a key characteristic of DeepMath-103K is its extensive topical diversity spanning the mathematical landscape. We categorized each problem using a hierarchical topic structure, following the methodology from Gao et al. (2024). As illustrated in Figure 4, this classification reveals that DeepMath-103K draws problems from a multitude of core mathematical areas. Its scope ranges from fundamental topics such as Prealgebra and Plane Geometry to sophisticated domains like Abstract Algebra (including Group Theory and Field Theory) and advanced Calculus (covering Differential Equations and Applications of Integrals, among others). This broad and deep topical foundation ensures that models trained on DeepMath-103K are exposed to a rich variety of mathematical concepts and problem-solving paradigms, thereby fostering the development of more robust and widely generalizable reasoning skills.

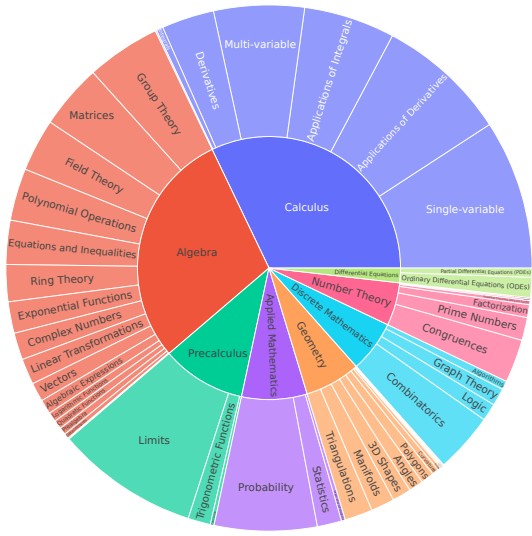

Figure 4: Hierarchical breakdown of covered mathematical topics in DeepMath-103K.

**Data Novelty and Uniqueness** As mentioned in § 1, DeepMath-103K sources mostly from math forum, rather than common resources frequently adopted by other datasets. To evaluate the data novelty and uniqueness of DeepMath-103K, we performed the following analysis for all the datasets:

1. We first embedded all the samples using paraphrase-multilingual-MiniLM-L12-v2.

2. Samples with an embedding similarity greater than 0.98 were considered as the same samples.

Viewing each dataset as a set of embeddings, Figure 5 presents the number of unique elements in each set and the corresponding set sizes. DeepMath-103K contains 82.81K problems that are not found in others. This stark contrast highlights the data novelty and uniqueness of DeepMath-103K. We also plot their embedding distribution after t-SNE in Figure 6. ORZ-129K (Hu et al.,

2025), Open-R1 (Face, 2025), SITLL-3-RL (Chen et al., 2025), DSR-Preview (Luo et al., 2025), and DAPO-17K (Yu et al., 2025), though curated independently, show very similar embedding distribution, while DeepMath-103K exhibits a distinctly different pattern. This observation supports our claim that existing datasets overlap with each other because of using common data sources and further demonstrate the data novelty and uniqueness of DeepMath-103K.

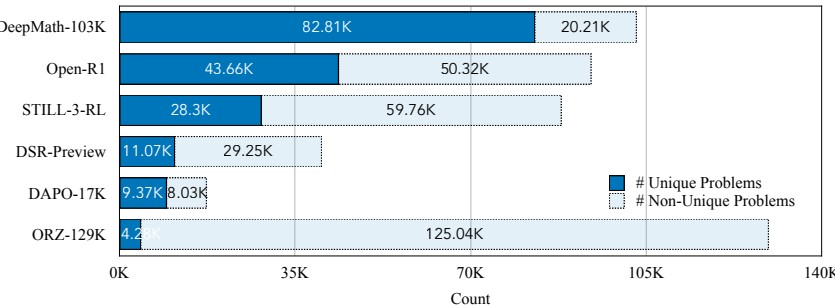

Figure 5: Unique and non-unique problem counts in DeepMath-103K compared to other datasets.

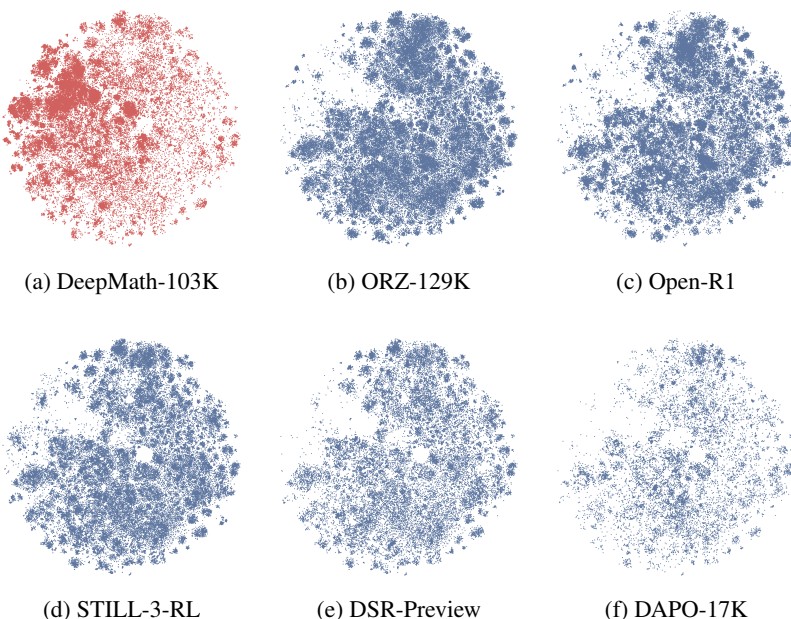

(a) DeepMath-103K    (b) ORZ-129K    (c) Open-R1

(d) STILL-3-RL    (e) DSR-Preview    (f) DAPO-17K

Figure 6: Embedding distributions of different datasets after t-SNE.

# 3 CONSTRUCTION OF DEEPMATH-103K

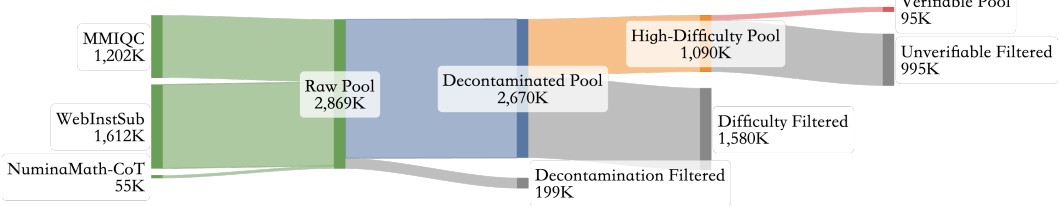

Figure 7: The data curation pipeline for DeepMath-103K. Starting with an initial pool of 2,869K raw questions, successive stages of data decontamination, difficulty filtering (retaining levels ⩾5), and answer verifiability filtering yield 95K problems. These are then combined with 8K problems from SimpleRL (Zeng et al., 2025b) to form the final DeepMath-103K dataset.

This section details the meticulous data curation process used to construct DeepMath-103K, illustrated in Figure 7. The process comprises four primary stages:

1. **Source Analysis and Collection:** Identifying and collecting mathematically challenging problems by analyzing the difficulty distributions of existing open data sources.

2. **Data Decontamination:** Rigorously decontaminating the collected data to remove potential overlaps with standard evaluation benchmarks, ensuring evaluation integrity.

3. **Difficulty Filtering:** Filtering the decontaminated problems based on difficulty, retaining only those assessed at level 5 or higher to focus on challenging content.

4. **Answer Verification:** Ensuring each curated problem possesses a verifiable final answer, consistently validated across multiple solution paths generated by DeepSeek-R1.

Overall, this curation pipeline ensures that DeepMath-103K is largely free from benchmark contamination and concentrates on challenging mathematical problems suitable for advanced reasoning model training.

**Stage 1: Source Analysis and Collection.** To identify data sources rich in challenging problems, we began by analyzing the landscape of existing open mathematical reasoning datasets designed for SFT. These datasets utilize diverse collection methods. For instance, datasets such as MetaMathQA (Yu et al., 2024), dart-math-hard (Tong et al., 2024), and OpenMathInstruct-2 (Toshniwal et al., 2024a) primarily focus on augmenting problems and solutions derived from established datasets like GSM8K (Cobbe et al., 2021) and MATH (Hendrycks et al., 2021b). In contrast, datasets like NuminaMath-CoT (LI et al., 2024), MMIQC (Liu et al., 2024), and WebInstruct-Sub (Yue et al., 2024) source content more broadly from the web, gathering materials such as exercises and discussions from online platforms (e.g., Math Stack Exchange). We follow Gao et al. (2024) to estimate the difficulty distributions of these potential source datasets, as shown in Figure 8, which reveals distinct patterns: datasets derived from GSM8K and MATH (MetaMathQA, dart-math-hard, OpenMathInstruct-2), along with NuminaMath-CoT, ex-

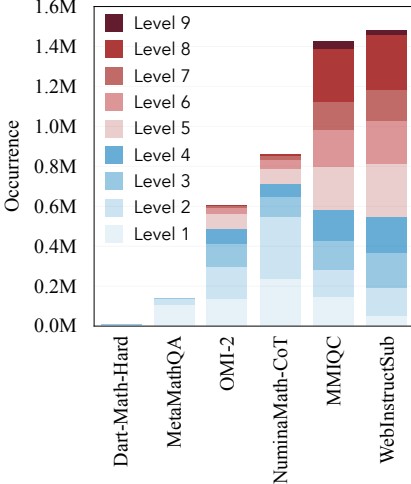

Figure 8: Difficulty distributions of various open mathematical datasets considered as potential sources.

hibited distributions heavily skewed towards lower difficulty levels (levels 1-5). Conversely, datasets sourced more broadly from web content, specifically MMIQC and WebInstructSub, displayed significantly flatter distributions with a larger proportion of problems in the mid-to-high difficulty range (levels 5-9). Based on this finding, we selected Math StackExchange subsets from MMIQC and WebInstructSub as our primary data sources due to their higher concentration of challenging problems. We also included NuminaMath-CoT to enhance the topical diversity of the initial collection. After applying basic filtering, this selection process yielded a raw pool of 2,869K questions.

**Stage 2: Data Decontamination.** As indicated by the high contamination rates observed in common benchmarks (Figure 3), a rigorous data decontamination process was crucial for ensuring the integrity of DeepMath-103K. We performed decontamination against a comprehensive suite of mathematics and STEM benchmarks, including MATH (Hendrycks et al., 2021b), AIME (MAA, a), AMC (MAA, b), Minerva Math (Lewkowycz et al., 2022), OlympiadBench (He et al., 2024), Omni-MATH (Gao et al., 2024), MathOdyssey (Fang et al., 2024), GAOKAO (Zhong et al., 2023), JEEBench (Arora et al., 2023), MMLU-STEM (Hendrycks et al., 2021a), CMATH (Wei et al., 2023), OlympicArena (Huang et al., 2024), GSM8K (Cobbe et al., 2021), and GPQA (Rein et al., 2024). We adopted the decontamination method proposed by Toshniwal et al. (2024a):

1. For each candidate question in our raw dataset, we employed embedding similarity search (using paraphrase-multilingual-MiniLM-L12-v2 (Reimers & Gurevych, 2019)) to identify the top-$k$ ($k = 5$) most similar examples from the aggregated test sets of all targeted benchmarks.

2. Each candidate question was then compared against its top-$k$ retrieved benchmark examples using an LLM-Judge (Llama-3.3-70B-Instruct (Grattafiori et al., 2024)) to determine if they

Table 1: Examples of contamination detected between the raw data pool and benchmarks using semantic comparison. Colors highlight conceptual or textual similarities.

| Benchmark | Raw Question | Benchmark Question |
|---|---|---|
| AIME24 | How many routes are there through from top left corner to bottom right in a 20x20 grid? I'm trying to solve this computer programming problem on Project Euler. I've seen a solution using nCr, where n = 40 and r = 20. Could someone explain to me how this work, please? | Consider the paths of length 16 that follow the lines from the lower left corner to the upper right corner on an 8x8 grid. Find the number of such paths that change direction exactly four times, as in the examples shown below. |
| AMC23 | Using only 3 paise, 5 paise, and 9 paise coins, what is the largest amount that cannot be paid in exact change? | In the state of Coinland, coins have values 6,10, and 15 cents. Suppose x is the value in cents of the most expensive item in Coinland that cannot be purchased using these coins with exact change. What is the sum of the digits of x? |

constituted identical questions or paraphrases. If any of these comparisons indicated a potential paraphrase or duplicate, the candidate question was discarded.

Table 1 illustrates the effectiveness of semantic decontamination compared to simple lexical matching. This approach aims to identify not only exact duplicates but also near-duplicates and paraphrased questions that might otherwise overlap with evaluation sets.

**Stage 3: Difficulty Filtering.** Zeng et al. (2025a) highlights the importance of aligning RL training data difficulty with the target model's reasoning capabilities, noting that powerful models benefit significantly from exposure to challenging problems. Building on this insight, our curation process for DeepMath-103K focuses on selecting problems that represent a significant reasoning challenge. To quantify difficulty, we adopted the approach detailed in Gao et al. (2024). We assigned a difficulty level to each decontaminated problem by prompting GPT-4o based on the annotation guidelines provided by the AoPS. To ensure a robust estimate, we queried GPT-4o six times for each problem and averaged the resulting ratings to determine its final difficulty level. We validated the consistency between GPT-4o's evaluation and human evaluation in Appendix D. Subsequently, we applied a strict filtering criterion, retaining only those problems with an estimated difficulty level of 5 or higher.

**Stage 4: Answer Verification.** The availability of verifiable final answers is crucial for enabling rule-based reward in RLVR, which helps mitigate reward hacking and has been instrumental in training successful reasoning models like DeepSeek-R1 (Guo et al., 2025). However, reliably constructing such answers presents two primary challenges:

1. Some open-ended questions inherently lack a easily verifiable final answer.

2. Certain answers are excessively complex (e.g., lengthy expressions or intricate notation), making them challenging or even infeasible for automated rule-based verification.

To address these issues, we implemented a rigorous two-stage verification process:

1. **Question Filtering and Formatting:** We used GPT-4o to process the raw questions. Problem types inherently unsuitable for verification were discarded. Questions phrased conversationally were rewritten into a standardized format seeking a single numerical or symbolic answer.

2. **Answer Verification via Consistency Check:** For questions successfully passing the above step, we generated three distinct solution paths using DeepSeek-R1. A rule-based verifier then extracted the final answer from each of these generated solutions, as well as from the original source solution (when available). We enforced strict consistency: only problems where all extracted final answers were identical were retained in the final dataset.

Question standardization and answer consistency checking ensures that every problem included in DeepMath-103K possesses a final answer that is robustly verifiable using automated rules.

Table 2: Math reasoning performance. "DeepMath" denotes models trained on DeepMath-103K.

| Model | MATH 500 | AMC 23 | Olympiad Bench | Minerva Math | AIME 24 | AIME 25 | Poly Math |
|---|---|---|---|---|---|---|---|
| *Zero RL from Base Model* | | | | | | | |
| Qwen-2.5-7B (Team, 2024) | 54.8 | 35.3 | 27.8 | 16.2 | 7.7 | 5.4 | 28.1 |
| ↳ Open-Reasoner-Zero-7B (Hu et al., 2025) | 81.8 | 58.9 | 47.9 | 38.4 | 15.6 | 14.4 | 40.7 |
| ↳ Qwen-2.5-7B-SRL-Zoo (Zeng et al., 2025a) | 77.0 | 55.8 | 41.0 | 41.2 | 15.6 | 8.7 | 33.1 |
| ↳ DeepMath-Zero-7B (Ours) | **85.5** | **64.7** | **51.0** | **45.3** | **20.4** | **17.5** | **42.7** |
| Qwen-2.5-Math-7B (Team, 2024) | 46.9 | 31.9 | 15.8 | 15.5 | 11.2 | 4.4 | 22.7 |
| ↳ Qwen-2.5-Math-7B-SRL-Zoo (Hu et al., 2025) | 75.8 | 59.7 | 37.4 | 29.9 | 24.0 | 10.2 | 36.0 |
| ↳ Oat-Zero-7B (Liu et al., 2025) | 80.0 | 66.7 | 43.4 | 40.8 | 32.7 | 11.7 | 40.8 |
| ↳ Eurus-2-7B-PRIME (Cui et al., 2025) | 80.2 | 64.7 | 44.9 | 42.1 | 19.0 | 12.7 | 38.9 |
| ↳ DeepMath-Zero-Math-7B (Ours) | **86.9** | **74.7** | **52.3** | **49.5** | **34.2** | **23.5** | **46.6** |
| Qwen-3-8B-Base (Team, 2025) | 60.3 | 42.5 | 33.1 | 27.2 | 11.2 | 7.9 | 30.0 |
| ↳ DeepMath-Zero-Qwen3-8B (Ours) | **93.2** | **86.2** | **66.3** | **53.4** | **44.6** | **33.5** | **52.3** |
| *RL from Instruct Models* | | | | | | | |
| R1-Distill-Qwen-1.5B (Guo et al., 2025) | 84.7 | 72.0 | 53.1 | 36.6 | 29.4 | 24.8 | 39.9 |
| ↳ DeepScaleR-1.5B-Preview (Luo et al., 2025) | 89.4 | 80.3 | 60.9 | 42.2 | **42.3** | 29.6 | **46.8** |
| ↳ Still-3-1.5B-Preview (Chen et al., 2025) | 86.6 | 75.8 | 55.7 | 38.7 | 30.8 | 24.6 | 43.1 |
| ↳ DeepMath-1.5B (Ours) | **89.9** | **82.3** | **61.8** | **42.5** | 37.3 | **30.8** | 46.6 |
| OpenMath-Nemotron-1.5B (Moshkov et al., 2025) | 91.8 | 90.5 | 70.3 | 26.3 | 61.3 | 50.6 | 56.8 |
| ↳ DeepMath-Omn-1.5B (Ours) | **93.2** | **94.2** | **73.4** | **28.3** | **64.0** | **57.3** | **58.7** |

# 4 DEEPMATH SERIES MODELS

This section presents a comprehensive evaluation of the mathematical and general reasoning capabilities of our DeepMath series of models, which were trained on DeepMath-103K.

**Training Paradigms**  We employed two distinct RL training paradigms:

- *Zero RL:* This paradigm involves training LLMs from their base (non-instruction-tuned) version using RL (Guo et al., 2025). We used group relative policy optimization (GRPO) (Shao et al., 2024) with fixes from Yu et al. (2025), and trained Qwen-2.5-(Math)-7B with a rule-based reward (+1 for correct final answer, -1 otherwise). Detailed settings are available in Appendix C.

- *RL:* We also performed RL on instruction-tuned models that already possessing math reasoning ability. We explored this using R1-Distill-Qwen-1.5B (Guo et al., 2025) and OpenMath-Nemotron-1.5B (Moshkov et al., 2025).

**Evaluation**  Following Zeng et al. (2025a;b), we assessed the mathematical performance of our models on: MATH-500 (Hendrycks et al., 2021b), AMC 2023 (MAA, b), OlympiadBench (He et al., 2024), Minerva Math (Lewkowycz et al., 2022), AIME 2024-2025 (MAA, a), and the English subset of PolyMath (Wang et al., 2025a). To investigate the generalization of reasoning abilities beyond mathematics, we used the GPQA-Diamond benchmark, which covers biology, physics and chemistry (Rein et al., 2024). For all evaluations, we adopted pass@1 accuracy (averaged over 16 samples) as the metric, and fixed the decoding parameters to temperature=0.6, top_p=0.95, and max_tokens=32K. To ensure a fair comparison and eliminate variance caused by the evaluation script, we re-evaluated the performance of all baseline models under our evaluation settings.

## 4.1 MATHEMATICAL REASONING RESULTS

The results presented in Table 2 collectively demonstrate the effectiveness of DeepMath-103K as a valuable resource for advancing the state-of-the-art in mathematical reasoning:

**Zero RL Training on Base Model**  DeepMath-Zero-7B and DeepMath-Zero-Math-7B, trained from the base Qwen-2.5-7B and Qwen-2.5-Math-7B models, demonstrate significant performance gains and surpass all baselines on evaluated benchmarks. These results highlight the effectiveness of DeepMath-103K in enabling the training of powerful reasoners from scratch.

**RL Training on Instruction-tuned Models**  Fine-tuning instruction-tuned models with RLVR on DeepMath-103K also yields notable performance enhancements. DeepMath-1.5B, initialized

from R1-Distill-Qwen-1.5B, achieves strong performance, particularly on AMC23 (82.3%) and OlympiadBench (61.8%). Similarly, DeepMath-Omn-1.5B, starting from OpenMath-Nemotron-1.5B, attains strongest results among 1.5B-scale models on all evaluated benchmarks. The consistent improvements observed across different instruction-tuned baselines further validate the utility of DeepMath-103K in boosting strong models.

**Ablation** Table 3 presents a head-to-head comparison with the representative open dataset ORZ-129K. Under the same RL recipe, training on DeepMath-103K alone outperforms training on ORZ-129K, while training on both achieves the highest performance. These results directly validate the effectiveness of our data and demonstrates that DeepMath-103K serves as a complement to existing resources (Figure 6). We also ablate the difficulty filtering step to verify its necessity. Removing this filter leads to a performance drop from 52.5% to 49.1%.

Table 3: Ablation results on training datasets and difficulty filtering strategy. Mean accuracy is computed across all evaluation sets in Table 2. Full results are listed in Appendix E.

| Model | Mean Acc. |
|---|---|
| Base Model (Qwen-2.5-Math-7B) | 21.2 |
| +ORZ-129K | 50.7 |
| +DeepMath-103K | 52.5 |
| −Difficulty Filtering | 49.1 |
| +Both (ORZ + DeepMath) | **53.0** |

## 4.2 GENERALIZABLE REASONING BEYOND MATHEMATICS

Table 4: Performance on the GPQA-Diamond, MMLU-STEM and BBH.

| Model | GPQA-Diamond | | | | MMLU-STEM | BBH |
|---|---|---|---|---|---|---|
| | Biology | Physics | Chemistry | Overall | | |
| *Zero RL from Base Model* | | | | | | |
| Qwen-2.5-7B | 33.6 | 27.8 | 21.4 | 25.3 | 10.8 | 41.3 |
| ↳ Open-Reasoner-Zero-7B | 50.3 | 47.8 | 26.7 | 38.1 | 61.1 | 82.8 |
| ↳ Qwen-2.5-7B-SimpleRL-Zoo | 31.9 | 37.9 | 22.6 | 30.2 | 51.6 | 71.3 |
| ↳ DeepMath-Zero-7B (Ours) | **57.2** | **53.0** | **28.2** | **41.7** | **72.7** | **84.8** |
| | | | | | | |
| Qwen-2.5-Math-7B | 32.2 | 26.0 | 21.1 | 24.3 | 19.3 | 35.9 |
| ↳ Qwen-2.5-Math-7B-SRL-Zoo | 40.1 | 31.2 | 22.9 | 28.2 | 34.8 | 58.6 |
| ↳ Oat-Zero-7B | **49.0** | 36.8 | 22.0 | 31.0 | 48.4 | 65.4 |
| ↳ Eurus-2-7B-PRIME | 44.1 | 37.4 | 24.1 | 31.8 | 40.4 | 58.2 |
| ↳ DeepMath-Zero-Math-7B (Ours) | 47.4 | **56.3** | **26.0** | **41.2** | **71.3** | **81.9** |
| | | | | | | |
| *RL from Instruct Models* | | | | | | |
| R1-Distill-Qwen-1.5B | 13.5 | 36.2 | 4.4 | 19.1 | 17.8 | 62.1 |
| ↳ DeepScaleR-1.5B-Preview | 15.5 | 46.8 | 9.1 | 26.1 | 19.5 | 65.9 |
| ↳ Still-3-1.5B-Preview | 16.8 | 38.4 | 5.2 | 20.7 | 24.1 | 65.8 |
| ↳ DeepMath-1.5B (Ours) | **18.1** | **47.6** | **12.2** | **28.2** | **33.0** | **69.1** |
| | | | | | | |
| OpenMath-Nemotron-1.5B | 12.8 | 23.5 | 18.9 | 20.3 | 41.3 | 47.4 |
| ↳ DeepMath-Omn-1.5B (Ours) | **17.1** | **28.4** | **21.5** | **24.1** | **48.6** | **54.0** |

Table 4 presents the performance of DeepMath models on GPQA-Diamond (biology, physics and chemistry) (Rein et al., 2024), MMLU-STEM (science and engineering reasoning) (Hendrycks et al., 2021a) and BBH (complex multi-step reasoning) (Suzgun et al., 2022) scores. DeepMath models achieve superior performance compared to other baselines, demonstrating a remarkable capacity to generalize their reasoning abilities acquired from math to broader domains. We attribute this generalization to the data diversity and rigorous curation. By sourcing less structured but more diverse data like Math StackExchange, DeepMath-103K yields a dataset with unique and diverse problems. Furthermore, the rigorous curation pipeline ensures both the challenge and the integrity of the data. This exposure to a wider variety of problem scenarios and reasoning styles likely equips our models with more robust and transferable reasoning skills.

## 4.3 ANALYSIS OF ZERO RL USING DEEPMATH-103K

Figure 9 analyzes the zero RL training. Specifically, Figure 9a illustrates the trend of response length throughout the training process, while Figure 9b tracks the emergence of four cognitive behaviors in Gandhi et al. (2025). The increasing trends in both response length and the manifestation of cognitive behaviors suggest a reproduction of the "aha moment" phenomenon observed in R1 (Guo

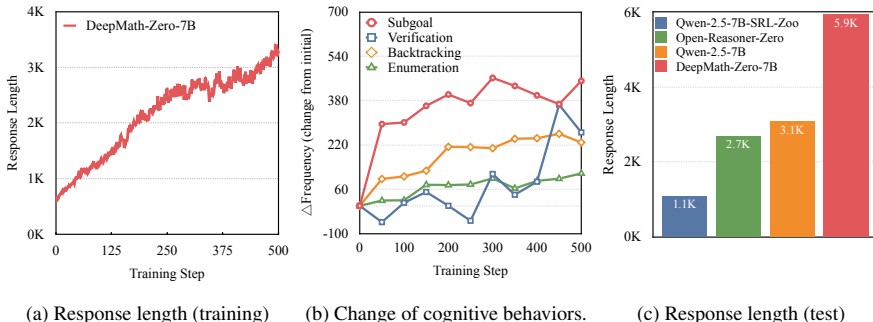

(a) Response length (training)    (b) Change of cognitive behaviors.    (c) Response length (test)

Figure 9: Training dynamics and response length of DeepMath-Zero-7B. (a) Rollout response length. (b) Cognitive behaviors. (c) Average response length on evaluated benchmarks.

et al., 2025). Furthermore, Figure 9c shows the average response lengths of different models on the evaluated benchmarks. The notably longer response lengths exhibited by DeepMath-Zero-7B suggest that more challenging problems can elicit deeper reasoning processes from the model.

## 5 CONCLUSION

In this work, we introduce DeepMath-103K, a large-scale mathematical dataset specifically designed to advance the reasoning capabilities of LLMs through RLVR. DeepMath-103K distinguishes itself through its high concentration of challenging problems, rigorous decontamination against a wide range of benchmarks, and the inclusion of verifiable final answers and multiple diverse solutions for each problem. Our data curation pipeline leverages the richness of less structured mathematical forums, resulting in a dataset with significant novelty and diversity compared to existing resources. Our experiments demonstrate the substantial impact of DeepMath-103K. Models trained on this dataset, the DeepMath series, achieve strong results on many mathematical benchmarks and exhibit remarkable generalization to domains beyond mathematics. By releasing the DeepMath-103K dataset, along with our code and model weights, we aim to provide a robust platform for the community to further explore and push the boundaries of advanced reasoning.

## 6 ETHICS STATEMENT

We, the authors of this work, confirm that we have read and adhered to the ICLR Code of Ethics. Our work focuses on the development of a large-scale mathematical dataset, DeepMath-103K, to advance reasoning in AI. We have taken great care in its construction, including a rigorous decontamination process, to ensure fair and trustworthy benchmark evaluations. The dataset and our associated training code will be made publicly available to promote open research and reproducibility.

## 7 REPRODUCIBILITY STATEMENT

To ensure the reproducibility of our work, we have made the DeepMath-103K dataset, along with all associated training code, scripts, and model weights, publicly available. Further details on our experimental setup, including hyperparameters and implementation specifics, are provided in Appendix C, enabling other researchers to fully reproduce and build upon our results.

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

## A    RELATED WORK

Datasets for advancing mathematical reasoning of LLM falls into three main lines corresponding to the three stages of LLM post-training: continue pre-training (CPT), SFT and RL. CPT aims to inject fundamental mathematical knowledge into LLMs with representative works like OpenWeb-Math (Paster et al., 2023), MathPile (Wang et al., 2024), InfiMM-Web-Math (Han et al., 2024), Fine-Math (Allal et al., 2025), and MegaMath (Zhou et al., 2025). SFT has been a foundational approach, utilizing datasets like MATH (Hendrycks et al., 2021b) and GSM8K (Cobbe et al., 2021) which provide problems with step-by-step solutions to teach models reasoning patterns. Subsequent efforts have focused on creating larger, harder and more diverse SFT datasets, such as MetaMathQA (Yu et al., 2024), OpenMathInstruct (Toshniwal et al., 2024b;a), NuminaMath-CoT (LI et al., 2024), MMIQC (Liu et al., 2024), dart-math-hard (Tong et al., 2024), and OpenMathReasoning (Moshkov et al., 2025). Recent progress in RLVR catalyzes datasets with verifiable reward, such as Open-R1 (Face, 2025), ORZ-129K (Hu et al., 2025), DSR-Preview (Luo et al., 2025), DAPO-17K (Yu et al., 2025), and BigMath (Albalak et al., 2025). DeepMath-103K distinguishes itself by a unique blend of high difficulty, rigorous decontamination, and verifiable answers.

## B    CONTAMINATION ANALYSIS OF EXISTING DATASETS

We performed a contamination analysis of several existing datasets, including ORZ-129K (Hu et al., 2025), DSR-Preview (Luo et al., 2025), DAPO-17K (Yu et al., 2025), Open-RS (Bansal et al., 2025), Open-R1 (Face, 2025), and DeepMath-103K. Our analysis focused on detecting potential contamination from the MATH500 (Hendrycks et al., 2021b), a commonly used benchmark. We employed a string-based comparison method, specifically identifying cases where the normalized in-del similarity between a problem in the analyzed dataset and a problem in MATH500 exceeded 90%. This approach is notably more lenient than the rigorous semantic decontamination procedure used in the construction of DeepMath-103K (§ 3). However, the numbers of contaminated samples shown in Figure 10 reveal that most of the analyzed datasets exhibit some degree of contamination, with the exception of DeepMath-103K.

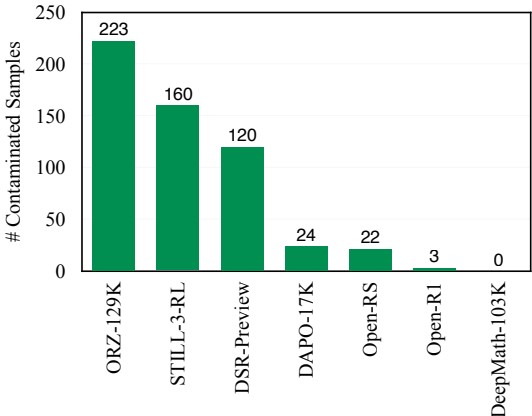

Figure 10: Number of contaminated samples in various datasets when compared against the MATH500 benchmark.

## C    TRAINING DETAILS

We use `verl` as the training framework[2]. Configurations are listed in Table 5.

Table 5: Configurations for training DeepMath series models.

| Config | DeepMath-Zero-7B | DeepMath-Zero-Math-7B | DeepMath-1.5B | DeepMath-Omn-1.5B |
|---|---|---|---|---|
| lr | 1e-6 | 1e-6 | 1e-6 | 1e-6 |
| kl_coef | 0.0 | 0.0 | 1e-3 | 1e-3 |
| max_prompt_length | 2K | 1K | 2K | 2K |
| max_response_length | 10K | 3K | 24K | 24K |
| train_batch_size | 512 | 512 | 128 | 128 |
| ppo_mini_batch_size | 32 | 32 | 64 | 64 |
| clip_ratio_low | 0.20 | 0.20 | 0.20 | 0.20 |
| clip_ratio_high | 0.28 | 0.28 | 0.27 | 0.27 |
| temperature | 1.0 | 1.0 | 0.6 | 0.6 |
| rollout_n | 16 | 16 | 16 | 18 |
| overlong_buffer.len | 2K | 512 | 4K | 4K |
| total_training_steps | 500 | 500 | 1800 | 700 |

---

[2]https://github.com/volcengine/verl

Table 6: Examples of geometry problems retained by the difficulty filtering process (level $\geqslant 5$).

| Difficulty | Problem |
|---|---|
| 5 | Four random points are placed in the plane, with each point's horizontal and vertical coordinates uniformly distributed on the interval $(0,1)$. What is the expected largest size of a subset of these points that can form the vertices of a convex polygon? |
| 6 | A square has one side lying on the line $y = 2x - 17$ and two other vertices on the parabola $y = x^2$. Determine the minimum possible area of the square." |
| 7 | Determine the sequence $s(k,n)$, which represents the number of sides of the intersection of a unit-radius regular polygon $P_k$ with $k$ sides and a rotating unit-radius regular polygon $P_n$ with $n \geqslant k$ sides, as the angle of rotation $\theta$ varies from 0 to $2\pi$. Provide the sequence $s(k,n)$ for all $n \geqslant k$. |
| 8 | Consider a convex n-gon $A_1 A_2 \cdots A_n$ inscribed in a unit circle. Determine the maximum value of the sum of the squares of all its sides and diagonals |
| 9 | Determine the maximal cardinality of a collection $\mathcal{C}$ of projective planes on $\omega$ such that no two distinct members of $\mathcal{C}$ are isomorphic. A set $L \subseteq \mathcal{P}(X)$ is a projective plane on $X \neq \varnothing$ if: 1. For any distinct $x, y \in X$, there is a unique $l \in L$ such that $x, y \in l$. 2. For any distinct $l, m \in L$, $|l \cap m| = 1$. 3. There exist four distinct elements of $X$ such that no member of $L$ contains more than two of these four elements. Two projective planes $L$ and $M$ on $X$ are isomorphic if there is a bijection $\varphi : X \to X$ such that $l \in L$ if and only if $\varphi(l) \in M$. |

## D  DIFFICULTY ASSESSMENT

As mentioned in § 3, our difficulty assessment follows the methodology from Omni-MATH (Gao et al., 2024), which relies on an automated, model-based evaluation. While the reliability of this approach was rigorously validated in Gao et al. (2024), we further verified the consistency between model and human judgment in Figure 11, which compares GPT-4o difficulty ratings with the established human-annotated difficulty levels from the MATH dataset (Hendrycks et al., 2021b) and shows a strong positive correlation. Table 6 also illustrates how increasing difficulty levels often correlate with greater conceptual depth and reasoning complexity, successfully capturing the inherent challenge of the problems.

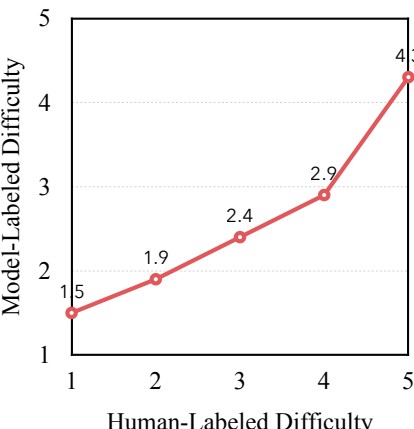

Figure 11: Correlation between model and human labeled difficulties.

## E  ABLATION

Table 7: Per-benchmark performance breakdown for Table 3.

| Model | MATH 500 | AMC 23 | Olympiad Bench | Minerva Math | AIME 24 | AIME 25 | Poly Math |
|---|---|---|---|---|---|---|---|
| Qwen-2.5-Math-7B | 46.9 | 31.9 | 15.8 | 15.5 | 11.2 | 4.4 | 22.7 |
| +ORZ-129K | 83.7 | 71.7 | 51.6 | 46.6 | 32.7 | 21.2 | 47.2 |
| +DeepMath-103K | 86.9 | 74.7 | **52.3** | 49.5 | **34.2** | 23.5 | 46.6 |
| −Difficulty Filtering | 84.5 | 70.9 | 48.6 | 46.8 | 31.5 | 16.0 | 45.2 |
| +Both | **87.5** | **75.0** | 51.7 | **50.9** | **34.2** | **24.0** | **48.0** |

## F    LLM USAGE STATEMENT

In the preparation of this manuscript, we utilized LLMs as an assistive tool. The primary applications of LLMs were to help polish the writing and enhance the clarity of the text, as well as to aid in various data processing tasks. The core research ideas, experimental design, and final analyses were conceived and executed by the human authors.

