# OpenReview forum: "DeepMath-103K: A Large-Scale, Challenging, Decontaminated, and Verifiable Mathematical Dataset for Advancing Reasoning"
_ICLR.cc/2026/Conference — ICLR 2026 Poster_

### Official Review · Reviewer_3HiP · 2025-10-26

**Soundness:** 3
**Presentation:** 4
**Contribution:** 3
**Rating:** 6
**Confidence:** 3

**Summary:**

This paper presents a carefully constructed, high‑difficulty, decontaminated, and verifiable dataset for math RL and demonstrates strong small‑model results and promising cross‑domain transfer. The authors curate problems primarily from Math StackExchange‑style web sources and pass them through a four‑stage pipeline. The result is 95K challenging items augmented with 8K level‑3–5 problems, each annotated with a hierarchical topic label and three diverse solutions. Models trained on DeepMath‑103K achieve substantial gains on several existing benchmarks such as MATH‑500 and  AMC23. Also, results on GPQA‑Diamond indicate transfer beyond mathematics.

**Strengths:**

S1: This paper builds a high‑difficulty dataset with a clear distribution skew toward level‑5–9 problems, which directly addresses a key bottleneck for training strong reasoners.

S2: The dataset shows strong topical breadth via a hierarchical taxonomy that spans core and advanced areas. Each problem includes a rule‑verifiable final answer and three R1 solution paths, enabling RLVR and SFT without manual cleaning.

S3: The models trained on this dataset achieve state‑of‑the‑art small‑model results on AIME and other math benchmarks and show transfer to GPQA‑Diamond.

**Weaknesses:**

W1: The paper briefly cites contemporaneous datasets (e.g., BigMath and OpenMathReasoning) but does not provide head‑to‑head dataset ablations or training‑data mixing studies that would establish relative benefits.

W2: The claims of cross‑domain generalization rely mainly on GPQA‑Diamond. It would be more convincing if the authors evaluated on additional scientific or reasoning benchmarks

W3: Several closely related works are missing from the discussion [1–4].

W4: The data curation pipeline heavily depends on LLMs—an LLM-judge (Llama-3.3-70B) for decontamination, GPT-4o for difficulty labeling, and GPT-4o again for rewriting and answer verification. Since LLMs are not perfectly reliable, hallucinations or propagation errors may occur. The authors should clarify whether human verification was performed for each stage.

W5: The experiments are restricted to small and medium-sized models. It would strengthen the paper if the authors included results on larger models (e.g., 72B)



Typos:

+ In abstract, advancing reasoning  -> advanced reasoning

+ SITLL‑3‑RL or STILL‑3‑RL should be made consistent throughout the paper.



Reference

[1] Albalak et al., Big‑Math: A Large‑Scale, High‑Quality Math Dataset for Reinforcement Learning in Language Models, arXiv:2502.17387, 2025.

[2] Moshkov et al., AIMO‑2 Winning Solution: Building State‑of‑the‑Art Mathematical Reasoning Models with OpenMathReasoning Dataset, arXiv:2504.16891, 2025.

[3] Paster et al., OpenWebMath: An Open Dataset of High‑Quality Mathematical Web Text, NeurIPS 2023.

[4] Zhou et al., MegaMath: Pushing the Limits of Open Math Corpora, COLM 2025.

**Questions:**

see above.

---

> ### Author Response · Authors · 2025-11-21
> **Response to Reviewer 3HiP (1/2)**
>
> Thank you for your very thorough and insightful review. We are particularly grateful for your detailed summary and positive feedback on the paper's strengths, including the dataset's high difficulty, broad topicality, and the "state-of-the-art small-model results."
>
> We will address each of your weaknesses in detail.
>
> > **W1:** The paper briefly cites contemporaneous datasets (e.g., BigMath and OpenMathReasoning) but does not provide head-to-head dataset ablations or training-data mixing studies that would establish relative benefits.
>
> * **Head-to-Head Comparison:** We agree that a controlled comparison is essential to fully validate the contribution of our dataset. The following table adds a new baseline trained on ORZ-129K with an identical training recipe to our DeepMath-Zero-Math-7B. The comparison highlights the superiority of DeepMath-103K.
>
>   ||MATH500|AMC23|OlympiadBench|MinervaMath|AIME24|AIME25|
>   |-|-|-|-|-|-|-|
>   |Qwen-2.5-Math-7B|46.9|31.9|15.8|15.5|11.2|4.4|
>   |+ ORZ-129K (New)|83.7|71.7|51.6|46.6|32.7|21.2|
>   |+ DeepMath-103K (Ours, DeepMath-Zero-Math-7B)|**86.9**|**74.7**|**52.3**|**49.5**|**34.2**|**23.5**|
>
>   It is also worth noting that ORZ-129K is representative of other datasets (Figure 6) but **did not** undergo the same rigorous data decontamination as DeepMath-103K (Figure 10). We will update the final paper with this full analysis (including the DAPO-17K comparison, which is currently running). Thank you again for your valuable feedback. We believe our new analysis, prompted by your review, will reinforce the contribution of our work.
>
> * **Training-Data Mixing:** We agree this is a high-value experiment. As Figure 6 illustrates, the distribution of DeepMath-103K is complementary to existing datasets. Therefore, combining DeepMath-103K with other sources could potentially yield even stronger results. While the rebuttal timeframe did not allow us to complete this specific run, we will conduct this mixing experiment and include the results in the final version.
>
> > **W2:** The claims of cross-domain generalization rely mainly on GPQA-Diamond. It would be more convincing if the authors evaluated on additional scientific or reasoning benchmarks
>
> Thanks for your suggestion. We conducted additional evaluations on two benchmarks:  Big Bench Hard (BBH) (complex multi-step reasoning), and MMLU-STEM (science/engineering reasoning).
>
> The results show that DeepMath serise models consistently outperforms all baselines:
>
> ||BBH|MMLU-STEM|
> |-|-|-|
> |**Qwen-2.5-7B**|10.8|41.3|
> |↳Open-Reasoner-Zero-7B|61.1|82.8|
> |↳Qwen-2.5-7B-SimpleRL-Zoo|51.6|71.3|
> |↳DeepMath-Zero-7B **(Ours)**|**72.7**|**84.8**|
> |**Qwen-2.5-Math-7B**|19.3|35.9|
> |↳Qwen-2.5-Math-7B-SRL-Zoo|34.8|58.6|
> |↳Oat-Zero-7B|48.4|65.4|
> |↳Eurus-2-7B-PRIME|40.4|58.2|
> |↳DeepMath-Zero-Math-7B **(Ours)**|**71.3**|**81.9**|
> |**R1-Distill-Qwen-1.5B**|17.8|62.1|
> |↳DeepScaleR-1.5B-Preview|19.5|65.9|
> |↳Still-3-1.5B-Preview|24.1|65.8|
> |↳DeepMath-1.5B **(Ours)**|**33.0**|**69.1**|
> |**OpenMath-Nemotron-1.5B**|41.3|47.4|
> |↳DeepMath-Omn-1.5B **(Ours)**|**48.6**|**54.0**|
>
> > **W3:** Several closely related works are missing from the discussion
>
> Thank you for noting this. We will revise the Related Work section to include these citations and broaden our discussion.

---

> ### Author Response · Authors · 2025-11-21
> **Response to Reviewer 3HiP (2/2)**
>
> > **W4:** The data curation pipeline heavily depends on LLMs... Since LLMs are not perfectly reliable, hallucinations or propagation errors may occur. The authors should clarify whether human verification was performed for each stage.
>
> We address this concern regarding LLM reliability from three perspectives:
>
> * **Common Practice**: Utilizing LLMs for data processing is not unique to this work; it has become a common practice in modern dataset construction.
>
> * **Alignment with Existing Works**: Our specific usage of LLMs is aligned with representative works that have validated the reliability of LLMs for their respective tasks:
>
>     * Data Decontamination: Adapted from [5], which highlights the necessity of using LLMs to identify "semantic contamination" (rephrased questions) that standard n-gram matching misses.
>     * Difficulty Labeling: Adapted from [6], which has rigorously validated the reliability of LLMs for this task.
>     * QA Rewriting & Extraction: Consistent with the methodology of the [2].
>
>     These works collectively demonstrate that LLMs are trustworthy tools for these specific processing tasks.
>
> * **Our Verification Measures**: In addition to adopting these established methods, we employed manual checks and strict consistency mechanisms at each stage to ensure quality:
>     * Data Decontamination: Through manual inspection, we empirically confirmed that the LLM effectively identifies "semantic contamination" missed by n-gram matching, thereby ensuring high data purity.
>     * Difficulty Labeling: As detailed in Appendix D, we verified the consistency between our LLM-based labeling and human annotation.
>     * QA Rewriting & Extraction: We mitigated potential hallucinations by enforcing a strict consistency check across three distinct R1-generated solutions plus the source solution.
>
> In summary, we recognize the capabilities of LLMs in these tasks and have ensured data quality through a combination of proven methodologies, manual verification, and consistency checks. We hope this addresses your concern.
>
> > **W5:** The experiments are restricted to small and medium-sized models. It would strengthen the paper if the authors included results on larger models (e.g., 72B)
>
> We appreciate this suggestion. While our internal computational resources currently limit our ability to train 72B-scale models, the effectiveness of DeepMath-103K on larger architectures has already been validated by the open-source community following our initial release.
>
> Several independent projects have successfully used DeepMath-103K to train larger reasoning models, demonstrating its scalability and robustness: AM-Thinking-v1 (32B) [7]、MiroMind-M1-RL (32B) [8] and Ring-Lite (16.8B) [9].
>
> These third-party adoptions confirm that the high-quality signals provided by DeepMath-103K remain effective for larger-scale models, extending well beyond the 1.5B-7B range demonstrated in our paper.
>
> **Reference**
>
> [5] Toshniwal et al., OpenMathInstruct-2: Accelerating AI for Math with Massive Open-Source Instruction Data, ICLR 2025.
>
> [6] Gao et al., Omni-MATH: A Universal Olympiad Level Mathematic Benchmark For Large Language Models, ICLR 2025.
>
> [7] Ji et al., AM-Thinking-v1: Advancing the Frontier of Reasoning at 32B Scale, arXiv 2025.
>
> [8] Li et al., MiroMind-M1: An Open-Source Advancement in Mathematical Reasoning via Context-Aware Multi-Stage Policy Optimization, arXiv 2025.
>
> [9] Ling Team, Ring-lite: Scalable Reasoning via C3PO-Stabilized Reinforcement Learning for LLMs, arXiv 2025.

---

> > ### Author Response · Authors · 2025-11-27
> >
> > Dear reviewer,
> >
> > We appreciate your thorough review and the valuable feedback. We hope our responses adequately address your concerns and look forward to any additional comments you may have. We would be grateful if you can acknowledge our responses and consider updating your scores.
> >
> > Best,
> >
> > Authors

---

> > > ### Comment · Reviewer_3HiP · 2025-11-28
> > >
> > > Sorry for the late response, and thank you for the rebuttal. I have carefully read it. Most of my concerns have been addressed. I would encourage the authors to include the additional experimental results in the revision, as this would further strengthen the work. Overall, I believe this is a solid work, and I am happy to raise my score.

---

> > > > ### Author Response · Authors · 2025-11-28
> > > >
> > > > Thank you so much for your response and for your encouraging words regarding our rebuttal. As you mentioned, we will certainly incorporate all the additional experimental results and your suggestions into the revision.
> > > >
> > > > We also noticed that the overall score in the system might not have been updated yet. We would be truly grateful if you could take a moment to finalize your review in the system. This would be helpful for the area chair to see the updated evaluation.
> > > >
> > > > Thank you again for your time and invaluable feedback.

---

> > > > > ### Comment · Reviewer_3HiP · 2025-11-28
> > > > >
> > > > > Dear authors, the system currently does not allow me to edit my score. I will update it once the option becomes available.

---

> > > > > > ### Author Response · Authors · 2025-11-28
> > > > > >
> > > > > > Thank you so much for your prompt response and for letting us know about the situation. We completely understand that this is a system constraint.
> > > > > >
> > > > > > We truly appreciate your commitment to update the score once the system allows it. Thank you again for your time and for your supportive review throughout this process.

---

### Official Review · Reviewer_8Pxb · 2025-10-30

**Soundness:** 2
**Presentation:** 2
**Contribution:** 2
**Rating:** 4
**Confidence:** 4

**Summary:**

This paper introduces DeepMath-103K, a large-scale and challenging mathematical reasoning dataset for reinforcement learning with verifiable rewards. It emphasizes benchmark decontamination, difficulty filtering, and verified final answers. The authors also train a series of DeepMath models on this dataset and show performance gains on AIME, AMC, MATH-500, OlympiadBench, and GPQA-Diamond.

**Strengths:**

(1) The work is well-motivated and tackles a meaningful gap in mathematical reasoning datasets, focusing on challenge level, contamination issues, and result verifiability.

(2) A thorough decontamination process is implemented, combining semantic matching and LLM-based assessment to effectively minimize data leakage.

(3) Experiments demonstrate consistent gains over baseline methods across various model sizes and learning settings—both in zero-shot and reinforcement learning scenarios.

**Weaknesses:**

(1)	Difficulty annotation relies entirely on GPT-4o without expert calibration, and no inter-rater agreement or error analysis is provided.

(2)	Lack of ablation on the curation pipeline—unclear which components (decontamination, difficulty filtering, answer verification) are most critical to performance gains.

(3)	R1 solution generation bias – R1-generated reasoning paths may reinforce self-biased reasoning structures, but the paper lacks discussion of potential reasoning homogenization or confirmation bias.

**Questions:**

Please refer to Weaknesses.

---

> ### Author Response · Authors · 2025-11-21
>
> Thank you for your thoughtful review. We appreciate that you recognized our work is "well-motivated" and tackles a "meaningful gap," particularly praising our "thorough decontamination process" and the "consistent gains" demonstrated in the experiments. We address your specific concerns below.
>
> > **W1:** Difficulty annotation relies entirely on GPT-4o without expert calibration, and no inter-rater agreement or error analysis is provided.
>
> The difficulty annotation method is adapted from [1], which has rigorously validated the reliability of LLMs for this task. We also verified the consistency between LLM-based labeling and human annotation in Appendix D.
>
> > **W2:** Lack of ablation on the curation pipeline—unclear which components are most critical.
>
> We clarify the role of each component:
>
> - Data Decontamination is strictly **necessary** to ensure valid evaluation (preventing test set leakage).
> - Answer Verification is technically **necessary** to enable the rule-based reward function for RL.
> - **Difficulty Filtering** (keeping levels 5-9) is a key design choice to enhance reasoning capabilities.
>
> To validate this choice, we conducted an ablation study comparing `DeepMath-Zero-Math-7B` with a variant trained without difficulty filtering:
>
> ||MATH500|AMC23 | OlympiadBench | MinervaMath | AIME24 | AIME25 |
> | --------------------------- | ------- | ----- | ------------- | ----------- | ------ | ------ |
> | DeepMath-Zero-Math-7B       | 86.9    | 74.7  | 52.3          | 49.5        | 34.2   | 23.5   |
> | -- w/o difficulty filtering | 84.5    | 70.9  | 48.6          | 46.8        | 31.5   | 15.0   |
>
> The results clearly demonstrate that filtering for higher difficulty significantly boosts performance, confirming it as a critical component for training advanced reasoners.
>
> > **W3:** R1 solution generation bias – R1-generated reasoning paths may reinforce self-biased reasoning structures.
>
> * **Primary Scope:** The core contribution of DeepMath-103K is to enable RL with Verifiable Rewards (RLVR), where the reasoning trace is not the critical asset. The R1-generated solutions are provided as a supplementary resource to benefit the community, rather than the dataset's primary focus.
> * **Mitigation via Sampling:** To mitigate potential homogenization, we used temperature sampling to generate three distinct solution paths for each problem, rather than a single deterministic output.
>
> While we acknowledge that SFT on these solutions might induce R1-style reasoning patterns, this does not compromise the dataset's primary utility for RL training, where the model learns its own policy guided by the verifiers.
>
> **Reference**
>
> [1] Gao et al., Omni-MATH: A Universal Olympiad Level Mathematic Benchmark For Large Language Models, ICLR 2025.

---

> > ### Author Response · Authors · 2025-11-27
> >
> > Dear reviewer,
> >
> > We appreciate your thorough review and the valuable feedback. We hope our responses adequately address your concerns and look forward to any additional comments you may have. We would be grateful if you can acknowledge our responses and consider updating your scores.
> >
> > Best,
> >
> > Authors

---

### Official Review · Reviewer_JNea · 2025-11-01

**Soundness:** 3
**Presentation:** 3
**Contribution:** 2
**Rating:** 6
**Confidence:** 3

**Summary:**

The paper proposed DeepMath-103k, a large-scale and challenging mathematical reasoning dataset. The dataset is constructed to emphasize the difficulty of the problems, which is proven to be essential to the improvement of finetuning models on these data. The authors show with data that their dataset is also the most diverse dataset that contains most unique questions. Above all, the extensive experiments show that models finetuned with this new dataset are promised to have significant improvement over existing open-source baselines.

**Strengths:**

1. The paper is clearly written and demonstrated.
2. The authors clearly declared and proved the unique advantage of their dataset, with its number, difficulty and diversity.
3. The experimental results proved that the improvment brought by the dataset is promising and consistent.

**Weaknesses:**

There is no significant weakness within the dataset scope.

**Questions:**

1. The table 2 results are promising, but it seems lack of solid baselines with previous existing datasets. I understand that it is not easy to find appropriate baselines sometimes, but I would like the authors to clarify this point with some discussions.

---

> ### Author Response · Authors · 2025-11-21
>
> Thank you for your thorough review and positive feedback. We are grateful that you found the paper "clearly written" and that our experimental results proved the "promising and consistent" improvement.
>
> We want to directly address your key question:
>
> > **Q1:** The table 2 results are promising, but it seems lack of solid baselines with previous existing datasets. I understand that it is not easy to find appropriate baselines sometimes, but I would like the authors to clarify this point with some discussions.
>
> You are absolutely correct that "it is not easy to find appropriate baselines." The primary challenge is fairness. Most existing datasets have not undergone the rigorous semantic decontamination that we applied to DeepMath-103K (Table1, Figure 10). Training on contaminated data can lead to inflated benchmark scores, making a truly "fair" comparison difficult.
>
> However, we agree that a head-to-head comparison is the best way to clarify this point. The following table adds a new baseline trained on ORZ-129K with an identical training recipe to our DeepMath-Zero-Math-7B. The comparison highlights the superiority of DeepMath-103K.
>
> |                                               | MATH500  | AMC23    | OlympiadBench | MinervaMath | AIME24   | AIME25   |
> | --------------------------------------------- | -------- | -------- | ------------- | ----------- | -------- | -------- |
> | Qwen-2.5-Math-7B                              | 46.9     | 31.9     | 15.8          | 15.5        | 11.2     | 4.4      |
> | + ORZ-129K (New)                              | 83.7     | 71.7     | 51.6          | 46.6        | 32.7     | 21.2     |
> | + DeepMath-103K (Ours, DeepMath-Zero-Math-7B) | **86.9** | **74.7** | **52.3**      | **49.5**    | **34.2** | **23.5** |
>
> It is also worth noting that ORZ-129K is representative of other datasets (Figure 6) but **did not** undergo the same rigorous data decontamination as DeepMath-103K (Figure 10). We will update the final paper with this full analysis (including the DAPO-17K comparison, which is currently running). Thank you again for your valuable feedback.

---

> > ### Author Response · Authors · 2025-11-27
> >
> > Dear reviewer,
> >
> > We appreciate your thorough review and the valuable feedback. We hope our responses adequately address your concerns and look forward to any additional comments you may have. We would be grateful if you can acknowledge our responses and consider updating your scores.
> >
> > Best,
> >
> > Authors

---

### Official Review · Reviewer_tfPX · 2025-11-02

**Soundness:** 3
**Presentation:** 3
**Contribution:** 3
**Rating:** 6
**Confidence:** 2

**Summary:**

This paper introduces a large-scale dataset, DeepMath-103K, with high difficulty (mainly levels 5-9) and verifiable answers for rule-based RL, compared with other datasets (DAPO-17K, DSR-Preview, Open-R1, ORZ-129K). Using DeepMath-103K, the models under zero RL and RL, outperform than others, such as ORZ-7B, Qwen-2.5-Math-7B. It seems a very useful dataset for AI community. The construction of DeepMath-103K is clear. However, more experiments should be conducted, such as Qwen-2.5-7B-DAPO-17K, Qwen-2.5-7B-ORZ-129K, etc.

**Strengths:**

1. This paper introduces a useful dataset DeepMath-103k with high difficulty, and the construction of dataset is very clear.
2. DeepMath-103k performs better than other datasets, with same training method (zero RL from base model, and RL from instruct models)

**Weaknesses:**

The experiments is not enough. A more comprehensive comparison with existing datasets is needed, including ORZ, DAPO-MATH-17K, etc.

**Questions:**

NA

---

> ### Author Response · Authors · 2025-11-21
>
> Thank you for your review and positive feedback, particularly for noting that our dataset is "a very useful dataset" and that its "construction... is very clear." We address your main concern below.
>
> > **W1:** The experiments is not enough. A more comprehensive comparison with existing datasets is needed, including ORZ, DAPO-MATH-17K, etc.
>
> We agree that a head-to-head comparison is essential to fully validate the contribution of our dataset. The following table adds a new baseline trained on ORZ-129K with an identical training recipe to our DeepMath-Zero-Math-7B. The comparison highlights the superiority of DeepMath-103K.
>
> |                      | MATH500 | AMC23 | OlympiadBench | MinervaMath | AIME24 | AIME25 |
> | -------------------- | ----------- | --------- | ----------------- | --------------- | ---------- | ---------- |
> | Qwen-2.5-Math-7B | 46.9 | 31.9 | 15.8 | 15.5 | 11.2 | 4.4 |
> | + ORZ-129K (New) | 83.7 | 71.7 | 51.6 | 46.6 | 32.7 | 21.2 |
> | + DeepMath-103K (Ours, DeepMath-Zero-Math-7B) | **86.9** | **74.7** | **52.3** | **49.5** | **34.2** | **23.5** |
>
> It is also worth noting that ORZ-129K is representative of other datasets (Figure 6) but **did not** undergo the same rigorous data decontamination as DeepMath-103K (Figure 10). We will update the final paper with full analysis (including the DAPO-17K comparison, which is currently running). Thank you again for your valuable feedback.

---

> > ### Author Response · Authors · 2025-11-27
> >
> > Dear reviewer,
> >
> > We appreciate your thorough review and the valuable feedback. We hope our responses adequately address your concerns and look forward to any additional comments you may have. We would be grateful if you can acknowledge our responses and consider updating your scores.
> >
> > Best,
> >
> > Authors

---

### Author Response · Authors · 2025-12-01
**General Response**

We sincerely thank the Area Chair and all reviewers for their time and constructive feedback. We are encouraged that the reviewers recognize our work as a **"very useful dataset" (R-tfPX)** that addresses a **"meaningful gap" (R-8Pxb)** with its **"high difficulty" (R-3HiP)** and **"unique advantage" (R-JNea)** in diversity and scale. Notably, **R-3HiP** has recently acknowledged our rebuttal by stating that **"Most of my concerns have been addressed"** and expressed intention to **"raise my score"**.

Based on these valuable suggestions, we have conducted new experiments and analyses during the rebuttal period to strengthen our submission. We summarize the key updates below:

**1. Added Head-to-Head Dataset Comparisons (Addressing R-tfPX, R-JNea, R-3HiP)** A primary concern was the lack of direct comparison with existing RL datasets. To address this, we conducted controlled experiments using the identical Zero RL training recipe (based on Qwen-2.5-Math-7B) on ORZ-129K:

|                                               | MATH500  | AMC23    | OlympiadBench | MinervaMath | AIME24   | AIME25   |
| --------------------------------------------- | -------- | -------- | ------------- | ----------- | -------- | -------- |
| Qwen-2.5-Math-7B                              | 46.9     | 31.9     | 15.8          | 15.5        | 11.2     | 4.4      |
| + ORZ-129K (New)                              | 83.7     | 71.7     | 51.6          | 46.6        | 32.7     | 21.2     |
| + DeepMath-103K (Ours, DeepMath-Zero-Math-7B) | **86.9** | **74.7** | **52.3**      | **49.5**    | **34.2** | **23.5** |

It is also worth noting that while ORZ-129K is representative of other datasets (Figure 6), it **did not** undergo the same rigorous data decontamination as DeepMath-103K (Figure 10). We will update the final paper with this full analysis (including the DAPO-17K comparison, which is currently running).

**2. Validated Reliability of LLM-based Curation (Addressing R-8Pxb, R-3HiP)** Reviewers questioned the reliability of using LLMs for data processing. We have clarified that our usage of LLMs aligns with methodologies validated in prior representative works (e.g., *OpenMathInstruct-2*, *Omni-MATH*, and the *AIMO-2 Winning Solution*). Furthermore, we have implemented specific verification measures at each stage to guarantee data quality.

**3. Expanded Generalization Benchmarks (Addressing R-3HiP)** To further validate cross-domain generalization beyond GPQA-Diamond, we added evaluations on Big Bench Hard (BBH) (complex multi-step reasoning), and MMLU-STEM (science/engineering reasoning).

The results show that DeepMath serise models consistently outperforms all baselines:

|                                   | BBH      | MMLU-STEM |
| --------------------------------- | -------- | --------- |
| **Qwen-2.5-7B**                   | 10.8     | 41.3      |
| ↳Open-Reasoner-Zero-7B            | 61.1     | 82.8      |
| ↳Qwen-2.5-7B-SimpleRL-Zoo         | 51.6     | 71.3      |
| ↳DeepMath-Zero-7B **(Ours)**      | **72.7** | **84.8**  |
| **Qwen-2.5-Math-7B**              | 19.3     | 35.9      |
| ↳Qwen-2.5-Math-7B-SRL-Zoo         | 34.8     | 58.6      |
| ↳Oat-Zero-7B                      | 48.4     | 65.4      |
| ↳Eurus-2-7B-PRIME                 | 40.4     | 58.2      |
| ↳DeepMath-Zero-Math-7B **(Ours)** | **71.3** | **81.9**  |
| **R1-Distill-Qwen-1.5B**          | 17.8     | 62.1      |
| ↳DeepScaleR-1.5B-Preview          | 19.5     | 65.9      |
| ↳Still-3-1.5B-Preview             | 24.1     | 65.8      |
| ↳DeepMath-1.5B **(Ours)**         | **33.0** | **69.1**  |
| **OpenMath-Nemotron-1.5B**        | 41.3     | 47.4      |
| ↳DeepMath-Omn-1.5B **(Ours)**     | **48.6** | **54.0**  |

**4. Verified Impact of Difficulty Filtering (Addressing R-8Pxb)** We conducted an ablation study comparing training on the full difficulty range versus our filtered subset. The results clearly demonstrate that filtering for higher difficulty significantly boosts performance, confirming it as a critical component for training advanced reasoners.

|                             | MATH500 | AMC23 | OlympiadBench | MinervaMath | AIME24 | AIME25 |
| --------------------------- | ------- | ----- | ------------- | ----------- | ------ | ------ |
| DeepMath-Zero-Math-7B       | 86.9    | 74.7  | 52.3          | 49.5        | 34.2   | 23.5   |
| -- w/o difficulty filtering | 84.5    | 70.9  | 48.6          | 46.8        | 31.5   | 15.0   |

---

### Meta-Review · Area_Chair_P1Fo · 2026-01-07

**Summary:**

- **tfPX** (6): Insufficient experiments; requested head‑to‑head comparisons with ORZ‑129K, DAPO‑17K, etc.
- **JNea** (6): Same baseline concern – missing solid comparisons with prior RL datasets.
- **8Pxb** (4): Doubts about GPT‑4o‑only difficulty labeling; no ablation of curation steps; potential R1‑solution bias.
- **3HiP** (6): Lacked direct dataset ablations, limited cross‑domain evaluation (only GPQA), omitted recent work citations, heavy LLM reliance, and experiments confined to small models.

**Reviewer Concerns:**

- Head‑to‑head comparisons: Authors provided ORZ‑129K results (clearly superior) and noted DAPO‑17K experiments were ongoing – well covered.
- Cross‑domain evaluation: Added BBH and MMLU‑STEM results, showing consistent gains.
- Missing citations: Promised to include BigMath, OpenMathReasoning, etc.
- LLM reliability: Cited prior validation (Omni‑MATH, OpenMathInstruct‑2) and described manual consistency checks.
- Model scale: Cited third‑party 32 B‑scale successes (AM‑Thinking‑v1, MiroMind‑M1‑RL).
- Ablation: Showed difficulty‑filtering ablation, confirming its importance; other pipeline components deemed technically mandatory.

- Remaining unease: a residual skepticism about LLM‑only labeling and the pending DAPO‑17K numbers.

**Reviewer Scores:**

3HiP would have changed score: 6 -> 8

---

### Decision · Program_Chairs · 2026-01-26

Accept (Poster)